# A Dilp8-dependent time window ensures tissue size adjustment in *Drosophila*

D. Blanco-Obregon [1], K. El Marzkioui[1], F. Brutscher[2], V. Kapoor[1], L. Valzania [1], D. S. Andersen[3,4], J. Colombani [3,4], S. Narasimha[1], D. McCusker[5], P. Léopold[1] & L. Boulan [1] ✉

The control of organ size mainly relies on precise autonomous growth programs. However, organ development is subject to random variations, called developmental noise, best revealed by the fluctuating asymmetry observed between bilateral organs. The developmental mechanisms ensuring bilateral symmetry in organ size are mostly unknown. In *Drosophila*, null mutations for the relaxin-like hormone Dilp8 increase wing fluctuating asymmetry, suggesting that Dilp8 plays a role in buffering developmental noise. Here we show that size adjustment of the wing primordia involves a peak of *dilp8* expression that takes place sharply at the end of juvenile growth. Wing size adjustment relies on a cross-organ communication involving the epidermis as the source of Dilp8. We identify ecdysone signaling as both the trigger for epidermal *dilp8* expression and its downstream target in the wing primordia, thereby establishing reciprocal hormonal feedback as a systemic mechanism, which controls organ size and bilateral symmetry in a narrow developmental time window.

One striking aspect of developmental processes is the precision with which final organ size is achieved and coordinated with other organs' dimensions to give rise to individuals with adequate proportions and functions. Although many developmental processes are now being characterized with great detail, the mechanisms of determination and fine adjustment of organ size are not understood correctly.

Symmetric bilateral organs constitute an ideal model for the study of developmental precision[1,2]. Since in most cases, left and right bilateral organs develop in the same environment, the limited, random asymmetries observed in adult bilateral traits reflect the stochastic variations taking place during development, also called developmental noise[3]. Developmental noise is generally quantified by the fluctuating asymmetry (FA) index, defined as the variance of the mean-scaled difference between left and right bilateral traits[1,2]. The low levels of variation observed between bilateral organs in physiological conditions suggest that buffering mechanisms are at play and maintain developmental robustness[4], although this is still under debate[5,6].

The identification of mutations affecting developmental precision in genetically tractable models opens the possibility to address such buffering mechanisms[7–10]. In *Drosophila*, mutations in the relaxin-like hormone Dilp8 and its receptor Lgr3 were recently found to decrease developmental stability through a systemic relay. Injured or tumorous imaginal discs produce a Dilp8 signal, which induces a delay in development and thereby allows for tissue repair. The steroid hormone ecdysone mediates this delay through a neural circuitry involving the Dilp8 receptor Lgr3[11–15]. In the absence of tissue injury, removing Dilp8 or Lgr3 function increases FA in adult wings, indicative of a physiological role for the Dilp8/Lgr3 axis in the control of developmental stability[11–13,15]. Although the mechanism of tissue repair-induced delay by Dilp8 is now better understood[16], the mechanism by which the Dilp8 hormone controls developmental

[1]Institut Curie, PSL Research University, CNRS UMR3215, INSERM U934, UPMC Paris-Sorbonne, 26 Rue d'Ulm, 75005 Paris, France. [2]Department of Molecular Life Sciences, University of Zurich, Winterthurerstrasse 190, CH-8057 Zurich, Switzerland. [3]Depatment of Biology, University of Copenhagen, Universitetsparken 15, 2100 Copenhagen, Denmark. [4]Novo Nordisk Foundation Center for Stem Cell Research, Faculty of Health and Medical Science, University of Copenhagen, Blegdamsvej 3B, 2200 Copenhagen N, Denmark. [5]University of Michigan, Ann Arbor, MI, USA. ✉e-mail: laura.boulan@curie.fr

stability in normal physiological conditions remains unknown. Moreover, the timing of size adjustment during physiological growth has not yet been quantified.

Two distinct hypotheses could account for such control. Continuous feedback taking place during the growth phase could maintain one organ on its growth trajectory, leading to an appropriate final size. In that case, a "robustness factor" would be expected to be produced by the organ itself as part of the feedback mechanism. Alternatively, developing organs could randomly deviate from a standard growth trajectory, up to a time window in development when the extent of the deviation is evaluated and a correction is made. If so, robustness factors could control either the emergence of the time window, the measure of the deviation, or its correction.

To address these issues, we first quantified the size variations of wing discs pairs at several time points during larval and early pupal development. We found that FA, while elevated during the growing larval stage, rapidly decreases after the larval-to-pupa (L/P) transition. In line with this finding, we observed that *dilp8* expression is sharply upregulated in the epidermis at the L/P transition and is functionally required for maintaining low FA. We also established that the burst of epidermal *dilp8* expression is directly controlled by the rise of ecdysone titer at the end of larval development. Finally, our results indicate that Dilp8 is in turn required to control the levels of ecdysone at the L/P transition, and that ecdysone receptor signaling is required in peripheral tissues for proper size adjustment.

We therefore propose a model whereby hormonal feedback between ecdysone and Dilp8 establishes a developmental time window early during pupal development when organ size is adjusted.

## Results

### A time window for wing imaginal disc size adjustment

As a first approach, we aimed to establish when the size of paired organs is adjusted by quantifying the left–right differences and FA in wing imaginal discs during development. To quantify the volume of the so-called wing "pouch" corresponding to the presumptive wing blade, we performed 3D reconstruction of the GFP-labeled *nubbin* expression domain (*nub > GFP*) of wing imaginal discs (Fig. 1a, see "Methods"). We plotted the left–right (L–R) volume difference for dissected pairs of discs at two time points during the larval phase (96 h after egg deposition (AED), corresponding to mid 3rd larval instar; and 114 h AED, late 3rd larval instar) and shortly after the larva-to-pupa (L/P) transition (7 h after puparium formation, or APF, after disc eversion is completed and dorsal and ventral sides of the pupal wings appose). In control *dilp8*[KO/+] heterozygous animals, which display adult FA comparable to wild type animals[17], the L–R variation of pouch volume is high at 96 h AED with an FA index (FAi) around 40 (Fig. 1b, d; black dots and bars). At 114 h AED, we observed a tendency toward a reduction of FA, although not statistically significant. A major adjustment then occurs between 114 h AED and 7 h APF, with an FAi dropping by 87% of its value at 96 h. The observation of a high FA during the larval phase, followed by a major correction around the L/P transition, favors the model of a time window for size adjustment.

In order to understand the role of Dilp8 in buffering FA and determine when size adjustment is lost in the absence of Dilp8 function, we performed the same analysis for *dilp8*[KO/KO] null mutants. In this genetic context, L–R variation was comparable to control during the larval phase, but remained high at 7 h APF (Fig. 1c, d; red dots and bars).

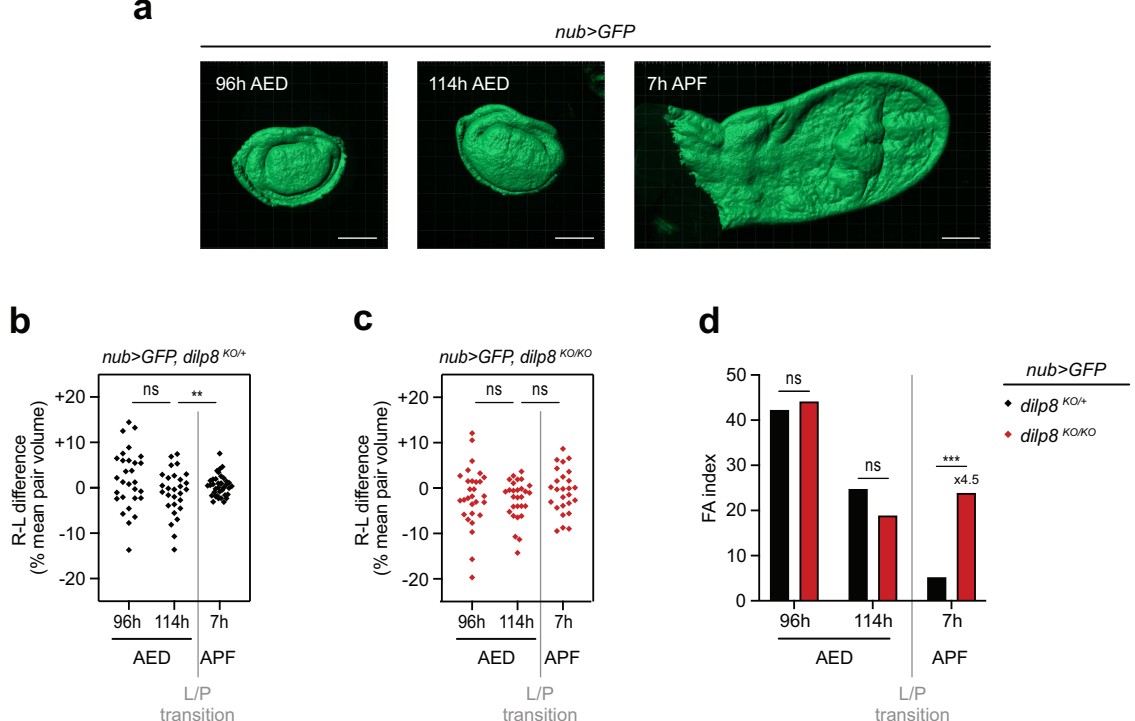

**Fig. 1 | A developmental time window for organ size adjustment.**
**a** Representative examples of surface reconstruction for volume measurements of the wing pouch domain labeled with *nub > GFP* at 96 h AED (mid L3 stage), 114 h AED (late L3 stage), and 7 h APF (early pupal stage). Scale bars represent 100 μm. These reconstructions were repeated each time for the n numbers indicated below. **b** Distribution of the right-left (R-L) pouch volume differences measured for individual pairs of wing discs and expressed as the percentage of the mean pair volume in control *dilp8* heterozygous animals (*nub > GFP, dilp8*[KO/+]). **p = 0.0012, Levene's test. **c** Distribution of the right-left (R-L) pouch volume differences measured for individual pairs of wing discs and expressed as the percentage of the mean pair volume in null *dilp8* mutant animals (*nub > GFP, dilp8*[KO/KO]). ns = not significant, Levene's test. **d** FA indexes calculated for each genotype and each timepoint based on the results shown in **b** and **c**. ***p = 0.0003 and ns = not significant, Levene's test. At 96 h AED, n = 29 pairs of discs from independent animals were analyzed for each genotype; at 114 h AED, n = 29 for each genotype; at 7 h APF, n = 37 for *dilp8*[KO/+], *nub > GFP* animals and n = 26 for *dilp8*[KO/KO], *nub > GFP* animals. AED after egg deposition, APF after pupa formation, L/P transition: larva-to-pupa transition. Source data are provided as a Source Data file.

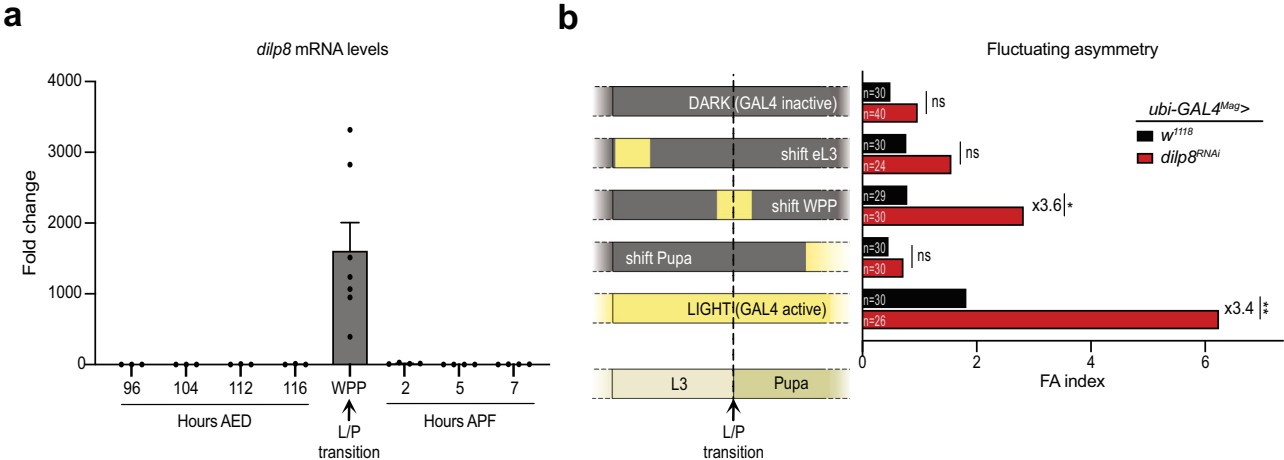

**Fig. 2 | A peak of Dilp8 expression at the L/P transition is required for organ size adjustment. a** Measurement of *dilp8* mRNA levels by qRT-PCR on whole animals (*w[1118]* as control strain) at the indicated time points during development. Values are expressed as fold changes relative to the 96 h AED timepoint. Error bars represent SEM. The white prepupa stage (WPP) corresponds to 0 h APF and marks the L/P transition. *n* = 3 biologically independent samples were analyzed for the 96–116 h AED time points, *n* = 7 for WPP and *n* = 4 for the 2–7 h APF stages. **b** Temporal downregulation of *dilp8* using the Shine-GAL4 system, in which the *ubi-GAL4[Mag]* driver is activated upon light exposure. The scheme on the left depicts the different light shifts. Yellow periods mark the developmental times at which animals were switched to white tube light to activate the GAL4/UAS system and downregulate *dilp8*. Chronic dark and light are the negative and positive controls, respectively. The graph on the right shows the FA indexes measured for adult pairs of wings of the given genotypes after the corresponding shift protocol. This experiment was performed at 29 °C. *n* values indicate the number of independent wing pairs analyzed; **$p$ = 0.0047, *$p$ = 0.0489 and ns = not significant, Levene's tests. AED after egg deposition, APF after pupa formation, L/P transition: larva-to-pupa transition, WPP white prepupa, eL3 early L3 stage. Source data are provided as a Source Data file.

Therefore, the difference in wing FA observed between *dilp8[KO/+]* and *dilp8[KO/KO]* animals (see Supplementary Fig. 1a) appears between late larval and early pupal stages. We conclude that Dilp8 is required for a major correction on wing disc size variation in a critical time window during development.

### A pulse of *dilp8* expression at the larva-to-pupa transition controls organ size adjustment

Given the role of *dilp8* in setting a time window for size adjustment at the L/P transition, we precisely analyzed the timing of its expression around this transition. Using qRT-PCR on carefully staged animals, we observed that *dilp8* expression is kept at low basal levels during larval development and is sharply upregulated at a stage called white prepupa (WPP) marking the end of larval stage (Fig. 2a). This dramatic increase (1500-fold) in *dilp8* mRNA accumulation drops within 2 h after WPP. We then used the Shine-GAL4 system (*ubi-GAL4[Mag]* driving *UAS-dilp8[RNAi]*), an optogenetic tool that allows conditional activation of the GAL4/UAS system using light exposure[18], to downregulate *dilp8* at different developmental stages and assess adult wing FA. Abrogation of the *dilp8* expression peak around WPP induced an increase in wings FA comparable to constitutive *dilp8* inhibition (Fig. 2b). As a control, inducing a light shift to silence *dilp8* earlier during the larval L3 stage, or later during pupal development, had no effect. Altogether, these results indicate that a peak of *dilp8* expression at WPP controls a time window for wing disc size adjustment taking place between WPP and 7 h APF.

### The larval epidermis is the source of Dilp8 for disc size adjustment

We next investigated the source of Dilp8 hormone responsible for size adjustment during early pupal development. In the context of perturbed disc growth, ill-growing discs autonomously produce Dilp8 and secrete it into the hemolymph. However, inhibiting *dilp8* expression specifically in wing discs did not increase adult wing FA (Supplementary Fig. 2a). This supports the notion that during normal growth, Dilp8-mediated size adjustment is not operating through a feedback mechanism where Dilp8 would be produced by the adjusting organ.

To identify the source of *dilp8* expression, qRT-PCR was performed on dissected tissues at the WPP stage. While very low or no expression was detected in wing discs, fat body, gut, brain and salivary glands, high *dilp8* expression was detected in the carcass, mainly composed of epidermis and muscles apposed together (Fig. 3a). Co-immunostaining with muscle and epidermal markers in the context of a *dilp8-GFP* transcriptional reporter (see "Methods" and[15]) indicated that *dilp8* is expressed specifically in the epidermis at the WPP stage (Fig. 3b, b'). This result was confirmed using a *dilp8-lacZ*[17] reporter construct (Supplementary Fig. 2b). In addition, the *dilp8-GFP* reporter showed a temporal upregulation at the WPP stage (Supplementary Fig. 2c), in accordance with our expression data on whole animals. To confirm the epidermal origin of Dilp8 at the WPP stage, we silenced *dilp8* expression using two epidermal drivers (*Eip71CD-GAL4* and *E22C-GAL4*) and two separate *UAS-dilp8[RNAi]* lines. In these conditions, the quantification of *dilp8* mRNA levels on whole animals showed an abrogation of the peak of *dilp8* expression at the WPP stage (Fig. 3c). By contrast, silencing *dilp8* with two muscle-specific drivers failed to suppress *dilp8* expression at WPP (Supplementary Fig. 2d), indicating that the larval epidermis is the unique source of Dilp8 hormone at the WPP stage.

Finally, we observed that silencing *dilp8* expression in epidermal cells is sufficient to induce adult wing FA (Fig. 3d), while downregulation of *dilp8* in the muscles does not affect developmental stability (Supplementary Fig. 2e).

Taken together, our results indicate that the epidermis is the source of a burst of *dilp8* expression at the WPP stage that triggers organ size adjustment.

### Ecdysone signaling triggers *dilp8* upregulation in epidermal cells at the larva-to-pupa transition

The sharp expression of *dilp8* in the WPP epidermis is indicative of a tight spatial and temporal transcriptional control. Temporally, ecdysone titers increase gradually during the L3 stage and reach maximum levels at the WPP stage[19]. Therefore, the peak of *dilp8* expression at WPP could rely on ecdysone. To test this possibility, we silenced expression of the *ecdysone receptor* (*EcR*) gene specifically in the epidermis using a weak RNAi line to prevent larval or early pupal lethality

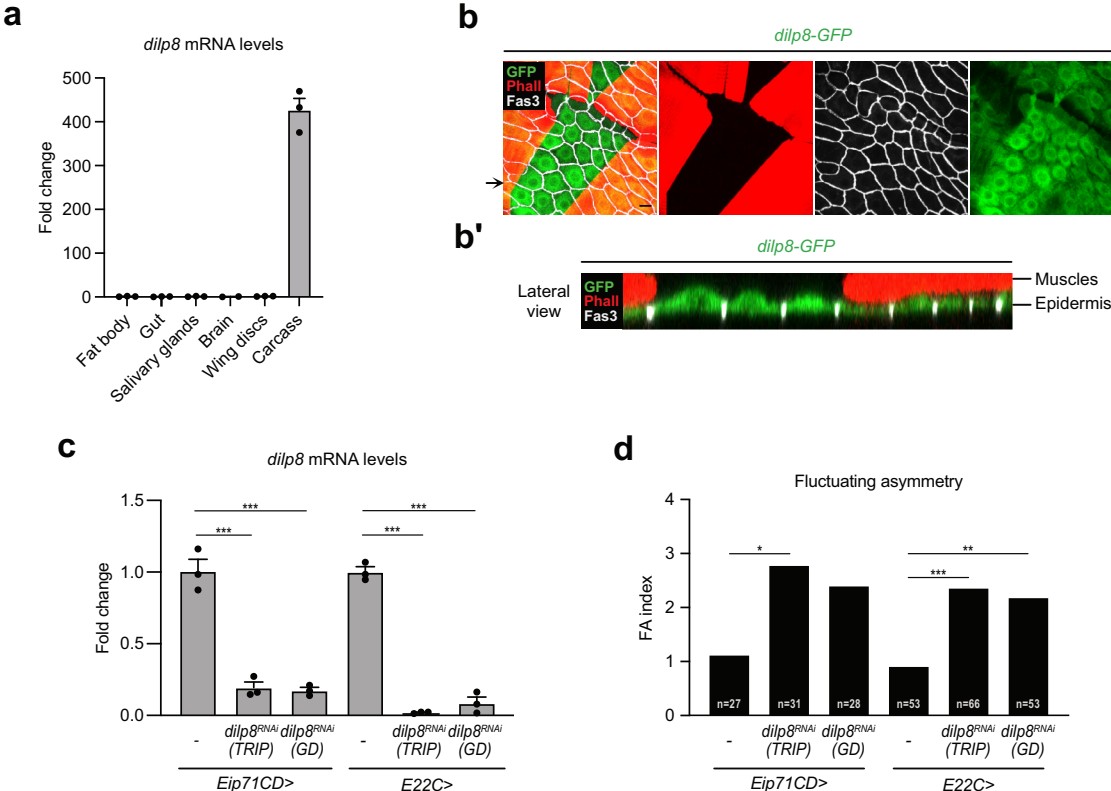

**Fig. 3 | The epidermis is the source of Dilp8 at the WPP stage. a** Measurement of *dilp8* mRNA levels by qRT-PCR on dissected tissues (*w[1118]* control strain) at the WPP stage. Values are expressed as fold changes relative to the fat body. Error bars represent SEM. *n* = 3 biologically independent samples were analyzed for all tissues, except for the brain where *n* = 2. **b** Maximal projection of a fillet preparation showing expression of the *dilp8-GFP* reporter at the WPP stage. Phalloidin (Phall, in red) staining marks actin filaments in the muscles and Fasciclin 3 (Fas3, in gray) marks the cell membranes of epidermal cells. The arrow on the side indicates the plane for lateral view reconstruction. Scale bar represents 20 μm. **b′** Lateral view of the preparation presented in **b**, showing that *dilp8-GFP* is expressed in the layer corresponding to epidermal cells and not in the muscles. This experiment was

repeated independently more than 3 times with similar results. Brightness and contrast adjustments were performed without altering signal localization. **c** Measurement of *dilp8* mRNA levels by qRT-PCR on whole animals at the WPP stage upon RNAi-mediated downregulation of *dilp8* in the epidermis. Values are expressed as fold changes relative to controls without RNAi. Error bars represent SEM. ***$p < 0.0001$, one-way ANOVA. *n* = 3 biologically independent samples were analyzed for all genotypes. **d** FA indexes of adult wings upon RNAi-mediated downregulation of *dilp8* in the epidermis. *n* values indicate the number of independent wing pairs analyzed; ***$p = 0.0004$, **$p = 0.0017$ and *$p = 0.0193$, for *Eip71CD* > *dilp8[RNAi]* GD, $p = 0.0522$; two-tailed *F*-tests. Experiments in **c** and **d** were done at 29 °C. WPP white prepupa. Source data are provided as a Source Data file.

(*Eip71CD* > *EcR[RNAi]pan* and *E22C* > *EcR[RNAi]pan*) and observed a strong decrease in *dilp8* expression at WPP (Fig. 4a). Interestingly, other pathways known to control *dilp8* expression in the context of tissue repair[14,16,17,20], like Hippo, JNK and Xrp1 signaling, are not required for epidermal *dilp8* expression at the WPP stage (Supplementary Fig. 3a, b). In addition, we analyzed epidermal cells at the WPP stage after clonal expression of a dominant-negative form of EcR (EcR[DN]), which binds ecdysone and the promoter region of target genes but is deficient for transcriptional activation. In EcR[DN]-expressing clones, the GFP signal corresponding to the *dilp8-GFP* reporter disappeared, in contrast with neighboring control cells (Fig. 4b). This confirmed the cell-autonomous control of *dilp8* expression by EcR signaling in epidermal cells.

In addition to these expression data, we investigated the role of EcR signaling upstream of Dilp8 in the control of developmental stability. Inhibiting EcR function in the epidermis (*Eip71CD* > *EcR-RNAi*, *E22C* > *EcR-RNAi*) led to a significant increase in adult wing FA (Fig. 4c). This establishes that ecdysone induces *dilp8* expression at WPP in the epidermis for the control of size adjustment.

### Dilp8 controls developmental precision through feedback on systemic ecdysone levels at the L/P transition

In conditions of tissue injury, Dilp8 delays development by inhibiting the peak of ecdysone that triggers the L/P transition[14,15]. To investigate

whether Dilp8 also acts upstream of ecdysone for organ size adjustment, we compared the levels of circulating ecdysone in control and *dilp8[KO/KO]* conditions at several time points around the L/P transition. We observed a modification of the peak of ecdysone in *dilp8[KO/KO]* animals, with a significant increase in circulating ecdysone at the WPP stage (0 h APF), followed by a sharper decrease between 2 and 8 h APF (Fig. 5a). Strikingly, the increase in ecdysone levels occurs precisely when *dilp8* expression peaks, suggesting that Dilp8 operates a fast and precise control on the intensity and timing of ecdysone accumulation. Importantly, we did not detect significant differences in ecdysone levels before the larva-to-pupa transition (Fig. 5a), suggesting that increased FA as observed in *dilp8[KO/KO]* mutants is not due to an acceleration of the L/P transition. To confirm this, we thoroughly measured pupariation time in *dilp8* loss-of-function conditions. Neither *dilp8[KO/KO]* nor *dilp8*-RNAi targeted to the epidermis induced a difference in pupariation time compared to controls (Supplementary Fig. 4a–c). Conversely, loss of function for the *PTTH* gene, a condition that delays the L/P transition[19], does not affect adult FA (Supplementary Fig. 4d). Thus, the duration of the larval period is not a key parameter for size adjustment.

Dilp8 controls developmental timing and ecdysone levels via a neuronal relay and its receptor Lgr3[11–13]. Recently, Heredia and colleagues identified a new role for Dilp8 in controlling behavior at pupariation[21]. This occurs via a cluster of Lgr3-positive neurons located

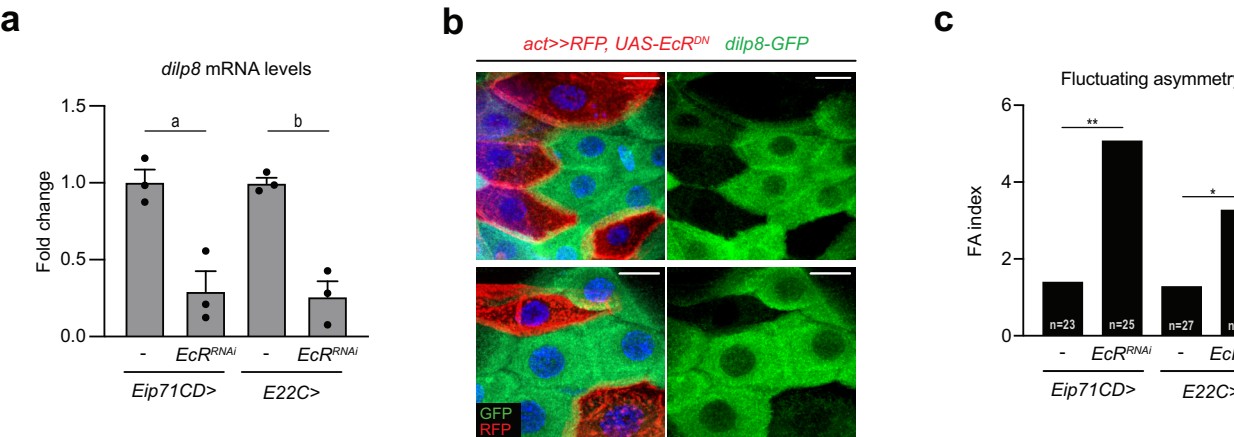

**Fig. 4 | Ecdysone signaling induces *dilp8* expression in the epidermis.**
**a** Measurement of *dilp8* mRNA levels by qRT-PCR on whole animals at the WPP stage upon RNAi-mediated downregulation of the ecdysone receptor gene *EcR* in the epidermis. Values are expressed as fold changes relative to controls without RNAi. Error bars represent SEM. a: $p = 0.0104$, b: $p = 0.0024$, two-tailed *t*-tests. $n = 3$ biologically independent samples were analyzed for all genotypes. **b** Two representative examples of epidermis of animals at the WPP stage showing expression of the *dilp8-GFP* reporter in control cells and clones of cells expressing a dominant-negative isoform of *EcR* (*EcR[DN]*; cells marked with RFP and shown in red). Scale bars represent 2 μm and DAPI stains the nuclei. This experiment was repeated independently 3 times with similar results. Brightness and contrast adjustments were performed equally on control and experimental conditions and without altering signal localization. **c** FA indexes of adult wings upon RNAi-mediated downregulation of *EcR* in the epidermis. *n* values indicate the number of independent wing pairs analyzed; \*\**p* = 0.0033 and \**p* = 0.0220, two-tailed *F*-tests. Experiments in **a** and **c** were done at 29 °C. WPP white prepupa. Source data are provided as a Source Data file.

in the ventral nerve cord. These neurons are marked by the *R18A01-GAL4* line, and appear to be distinct from the set of Lgr3 neurons that controls ecdysone levels (marked by the *R19B09-GAL4* line)[11–13]. We observed that the downregulation of Lgr3 with *R19B09-GAL4*, but not with *R18A01-GAL4*, increases FA (Supplementary Fig. 4e), suggesting that Dilp8 acts on FA via a brain circuitry that controls ecdysone levels. Indeed, reducing ecdysone production in *dilp8* loss of function by downregulating *PTTH* expression was sufficient to rescue wing FA (Supplementary Fig. 4f). Altogether, these data indicate that the level of ecdysone at the L/P transition is a key parameter for size adjustment.

**Dilp8 controls developmental precision by modulating ecdysone signaling in target tissues**

Since systemic ecdysone levels control FA, we investigated the possibility that ecdysone acts directly on wing disc to control specific parameters of tissue growth. We first assessed whether ecdysone signaling is modified in target tissues in absence of Dilp8. For this, we compared the expression levels of EcR target genes in dissected wing imaginal discs from *dilp8[KO/KO]* and control animals. We observed that 8 out of 9 selected EcR target genes were significantly upregulated in wing discs in the absence of Dilp8 (Fig. 5b), indicating a clear effect on the level of EcR signaling in target tissues at WPP.

We then tested whether these changes in ecdysone signaling in wing discs are causal for the elevated FA observed in *dilp8* null mutants. In the context of *dilp8[KO/+]* controls and *dilp8[KO/ag54]* null mutants, we downregulated the ecdysone importer EcI specifically in wing discs (*UAS-EcI[RNAi]* driven by *nub-GAL4* combined with an *elav-GAL80* to prevent expression in the brain), thereby reducing ecdysone signaling in target tissues. In these conditions, we observed a full rescue of the elevated wing FA observed in *dilp8[KO/ag54]* mutants (Fig. 5c), demonstrating that Dilp8 acts by ensuring proper ecdysone signaling in target tissues at the WPP stage.

Ecdysone signaling induces a G2 cell cycle arrest in wing discs between 2 and 6 h APF[22]. By quantifying cell proliferation in the wing pouch of control and *dilp8[KO/KO]* animals at and after the L/P transition, we observed a significant reduction in PH3-positive cells at 2 h APF in *dilp8[KO/KO]* wing discs (Fig. 5d), consistent with the increase in ecdysone signaling observed at WPP. Therefore, *dilp8[KO/KO]* discs experience a precocious cell cycle arrest during early prepupal development, in line with a slight reduction in adult wing size (Supplementary Fig. 1 and[15,17]). Overall, this data suggests a possible link between the total number of cells and the precision of pupal wing size adjustment.

## Discussion

Precision and stability are fundamental properties of many developmental processes, albeit poorly understood. Paired symmetrical organs have proven useful to quantify stochastic variation of developmental processes. However, studies on how organisms establish bilateral organ symmetry have been limited by the difficulty in precisely quantifying 3D morphogenesis on both sides of developing organisms. We provide here the first evaluation of bilateral wing disc development in *Drosophila*.

By measuring the volume of wing precursors and quantifying L−R variation, we find that wing discs undergo a major adjustment step during a developmental time window, which relies on the relaxin-like hormone Dilp8. Interestingly, a time window has also been described in zebrafish development during which bilateral inner ears and somites adjust their size[23,24].

In *Drosophila*, several important events for wing development take place during the first hours of the prepupa stage, among which are disc eversion and tissue expansion[25]. The adjustment of zebrafish inner ears also takes place in parallel to an important tissue expansion[23], suggesting that this process could be a general requirement for the adjustment of bilateral organs.

These findings contrast with the mechanism by which Dilp8 induces a developmental delay following alterations of disc growth. In this case, Dilp8 is produced by ill-growing tissues and triggers a feedback mechanism on ecdysone production, allowing coupling the growing state of organs with the major developmental transition at the end of the juvenile period. We show here that in absence of perturbation, Dilp8 is produced in a specific tissue, the epidermis, and is required at a precise stage to ensure organ size adjustment.

We demonstrate reciprocal feedback between ecdysone and Dilp8 taking place at the WPP stage: while ecdysone is needed for *dilp8* expression, Dilp8 feeds back on ecdysone production and adjusts its

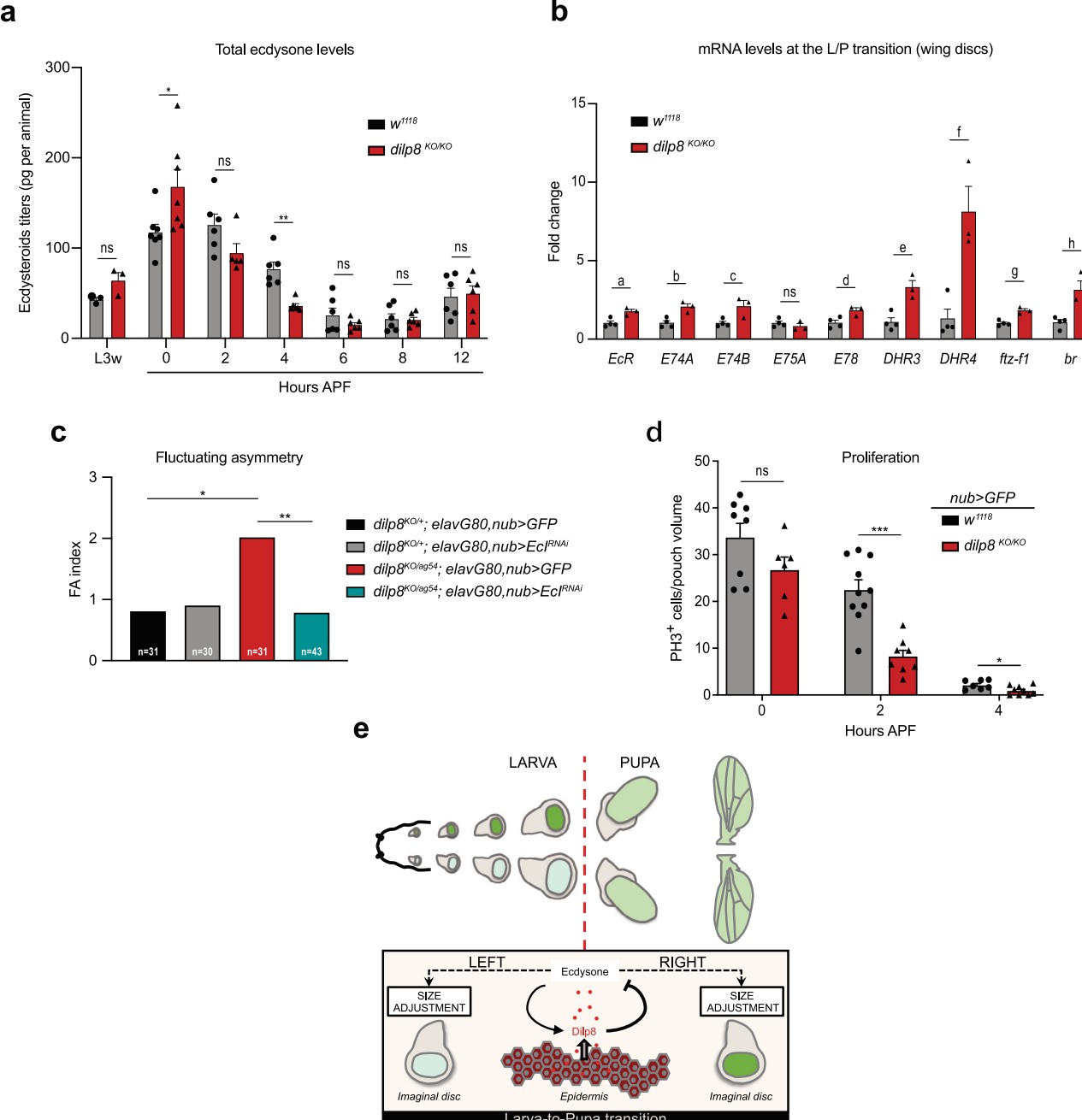

**Fig. 5 | Dilp8 is required to adjust ecdysone levels at the L/P transition.**
**a** Ecdysteroids titers in whole animals at the indicated time points for controls (in black) and *dilp8^KO/KO* (in red) mutant animals. At L3w, *n* = 3 biologically independent samples measured per genotype; at 0 h AFP, *n* = 7 per genotype; for other time points, n = 6 per genotype except for dilp8^KO/KO 2 h APF, where *n* = 5. Error bars represent SEM. **\*\****p* = 0.0006, *\*p* = 0.0338 and ns = not significant, multiple two-tailed *t*-tests. **b** Measurement of ecdysone targets by qRT-PCR on wing imaginal discs at WPP stage in controls (in black) and *dilp8^KO/KO* (in red) mutant animals. Values are fold changes relative to controls. Error bars represent SEM. a: *p* value = 0.0119, b: *p* value = 0.0076, c: *p* value = 0.0253, d: *p* value = 0.0248, e: *p* value = 0.0058, f: *p* value = 0.0069, g: *p* value = 0.0019, h: *p value* = 0.0114 and ns = not significant, multiple two-tailed *t*-tests. *n* = 4 biologically independent samples analyzed for controls and *n* = 3 for *dilp8^KO/KO* animals. **c** FA indexes of adult

wings of the indicated genotypes, showing that *dilp8* loss of function can be rescued by specifically downregulating ecdysone signaling in the wing discs. *n* values indicate the number of independent wing pairs analyzed. *\*p* = 0.0145 and **\*\****p* = 0.0049, two-tailed *F*-tests. **d** Proliferating cells relative to the wing pouch volume (*nub > GFP* domain) of control *dilp8^KO/+* and homozygous *dilp8^KO/KO* animals. Error bars represent SEM. **\*\*\****p* = 0.0001, *\*p* = 0.0303, ns = not significant, two-tailed *t*-tests. For controls, *n* = 8 wing discs from independent animals analyzed at 0 h APF, *n* = 10 at 2 h APF and *n* = 7 at 4 h APF. For *dilp8^KO/KO* animals, *n* = 6 at 0 h APF, *n* = 8 at 2 h APF and *n* = 9 at 4 h APF. **e** Schematic model on the reciprocal interaction between epidermal Dilp8 and Ecdysone signaling required at the larva-to-pupa transition for proper wing disc size adjustment. L3w: wandering late L3 stage, APF after pupa formation, L/P transition: larva-to-pupa transition. Source data are provided as a Source Data file.

levels of signaling in target tissues (see our model in Fig. 5e). We show that none of the upstream signals needed for Dilp8 induction in response to tissue stress (i.e. JNK, Xrp1) is needed for its developmental expression. Although previously shown to contribute to *dilp8*

expression[17], the transcriptional activator Yorkie/Scalloped does not participate in physiological *dilp8* induction at the WPP. Noticeably, removing a Yki/Sd response element in the *dilp8* promoter leads to a rather mild decrease in developmental stability[17]. This indicates that

Yki/Sd plays a limited role in Dilp8-dependent developmental stability, distinct from the major regulation step occurring at WPP.

Expression of *dilp8* in the larval epidermis indicates that epidermal cells play a major endocrine role at the larva-to-pupa transition. Intriguingly, while most larval tissues express *EcR* and *Usp* and respond to ecdysone during this critical transition, only epidermal cells contribute to EcR-dependent *dilp8* expression. This could result from the functional preponderance of specific co-factors present in the epidermis possibly required together with EcR for *dilp8* induction at WPP. Taiman (Tai) is a co-factor of EcR required for the induction of *dilp8* in wing discs overexpressing Yki[26], but only a minor decrease in *dilp8* expression at WPP was observed after silencing *tai* in epidermal cells (Supplementary Fig. 3b).

The larval epidermis undergoes important transformations at the WPP. A series of contractions shorten the size of the future body and the cuticle sclerotizes to produce a rigid pupal case. These events follow a precisely staged sequence allowing progression into pupal development. Interestingly, epidermally produced Dilp8 is required for proper accomplishment of this complex behavioral series[21]. Therefore, the production of Dilp8 from the epidermis could allow an integration of major morphological and timer functions needed at the L/P transition.

A parallel should be made between emerging endocrine properties of the larval epidermis presented here and in[21], and the established neuroendocrine functions of the vertebrate skin. Cutaneous structures respond to, but also generate, a large number of neuromodulators and hormones, which participate in skin homeostatic functions including metabolic activity, tissue repair, immune response (for review[27]). Several neuropeptides were identified from amphibian skin before being found in neural tissues, and human skin recapitulates the TRH/TSH/thyroid and the CRH/ACTH/Cortisol axes found in the central brain[28,29]. These observations have suggested an ancestral function for the epidermis as a neuroendocrine organ. In this context, our finding of a Dilp8 epidermal function suggests possible conserved cross-talks between neurohormonal brain and epidermal axes.

We and others previously showed that Dilp8 acts through a limited number of Lgr3-positive neurons in the central lobe region of the larval brain to delay ecdysone production and the L/P transition in response to growth impairment[11–13]. Interestingly, depleting Lgr3 in this subpopulation of neurons promotes high adult wing FA, suggesting that Dilp8 controls developmental stability through the same neuronal relay.

Our present findings indicate that the temporal accumulation of ecdysone is modified in *dilp8* loss-of-function conditions. Consistently, higher expression levels of EcR targets are found in wing discs at WPP. Collectively, these and our previous results indicate that epidermal Dilp8 acts on ecdysone accumulation though an Lgr3 central relay and modulates EcR signaling in peripheral tissues for disc size adjustment. Interestingly, Dilp8 is a temporal neuromodulator of ecdysone function, both in the context of growth impairment and during normal development. The sharp induction of *dilp8* at WPP and the immediate response on ecdysone levels observed upon *dilp8* loss of function (see Fig. 5a, b) indicate that Dilp8-mediated neurohormonal action has high temporal definition, an important property for its function as a developmental timer.

In conclusion, our results define a hormonal crosstalk between ecdysone and Dilp8 with two functions: (i) it defines a time window after larval development during which wing disc size is adjusted; (ii) it allows a fine-tuning of ecdysone signaling in the discs, which appears crucial for size adjustment. The dynamics of ecdysone levels in early pupal wing discs controls a transcription cascade leading to two waves of cell division/cell cycle exit[22]. Further work will be needed to understand how the systemic ecdysone signal contributes to adjusting organ size during pupal development.

## Methods

### Fly strains and food

The following RNAi lines were obtained from the Vienna *Drosophila* RNAi Center (VDRC): *UAS-yki^{RNAi}* (KK 104523), *UAS-sd^{RNAi}* (KK 108877), *UAS-dilp8^{RNAi}* (GD 9420), *UAS-Ecl^{RNAi}* (GD 37295) and *UAS-ptth^{RNAi}* (KK 102043). The *w^{1118}* (BL 3605), *UAS-GFP* (BL 35786), *UAS-dilp8^{RNAi}* TRIP (BL 80436), *dilp8-GFP* (*dilp8^{MI00727}*; BL 33079), *tub-GAL80^{ts}; tub-GAL4* (BL 86328), *mef2-Gal4* (BL 27390), *mhc-GAL4* (BL 84298), *UAS-xrp1^{RNAi}* (BL 34521), *UAS-bsk^{DN}* (BL 6409), *UAS-EcR^{DN}* (F645A, BL 6869), *UAS-EcR^{RNAi}pan* (BL 29374), *UAS-jub^{RNAi}* (BL 32923), *Eip71CD-GAL4* (BL 6871), *act^{SC}-GAL4* (BL 4414), *R19BO9-GAL4* (BL 48840), *R18AO1-GAL4* (BL 48791), *UAS-lgr3^{RNAi}* (BL 36887) lines were provided by the Bloomington *Drosophila* Stock Center. Other lines used in this study were: *ubi-GAL4^{Mag 18}; nub-GAL4^{30}; dilp8^{ag54 21}; dilp8^{KO}* and *dilp8-full-prom-lacZ^{17}; ptth* mutant[19]; *E22C-GAL4^{31}; hs-flp; act-FRT-STOP-FRT-GAL4,tub-GAL80^{ts}; UAS-RFP* (gift from the Bellaïche Lab).

Animals were reared at 25 °C (unless otherwise stated in the figure legends) on fly food containing, per liter: 14 g inactivated yeast powder, 69 g corn meal, 7.5 g agar, 52 g white sugar, and 1.4 g Methyl 4-hydroxybenzoate.

### Light-shift experiments using the Shine-GAL4 system

Crosses with the *ubi-GAL4^{Mag}* line were left for egg laying during 8 h on plates made of 2% agar and 2% sucrose in PBS. The next day, synchronized L1-stage animals were transferred to vials with fly food and kept in the dark to repress GAL4 activity until the indicated times. At this point, the tubes with synchronized larvae were shifted to ambient light to quickly allow GAL4 activation[18].

The experiment with *UAS-dilp8^{RNAi}* presented in Fig. 2b was performed at 29 °C. For the shift at the early L3 stage, animals were exposed to light at 72 h AED for 24 h. For the shift covering the WPP stage, animals were exposed to light at 96 h AED for 24 h, after which nearly all animals had pupariated (remaining larvae were removed). For the shift at the pupal stage, animals were exposed to light at 24 h APF until adulthood.

### Measurement of the FA index

In the case of wing primordia measurements (Fig. 1), female flies were dissected at the given time points, fixed in 4% formaldehyde (Sigma) in PBS for 30 min at room temperature, and washed in PBS. For early pupae (7 h APF), animals were dissected as described in ref. 32. The left- and right-wing discs of each individual were mounted without coverslip on "Cellview" cell culture dishes with glass bottom (Greiner Bio-one, #627861), in order to preserve the original structure volume. Imaging was performed with a Zeiss LSM900 Inverted Laser Scanning Confocal Microscope using identical settings for each pair. The confocal Z-stacks were processed with the Imaris software using identical settings for each pair, and surfaces of the *nub > GFP* signal were generated to faithfully represent the original structure volume.

In the case of adult wing measurements, adult female flies of the appropriate genotypes were collected, stored in ethanol and mounted in a lactic acid: ethanol (6:5) solution. Wings were dissected and mounted in pairs. Pictures were acquired with a 1024 ×768 resolution using a MZ16-FA Leica Fluorescence Stereomicroscope with a DFC-490 Leica digital camera (Bright-field mode, 50% illumination intensity, 10.5 exposure, 2.3 gain, 152 saturation and 1.20 gamma).

We used the FA index (FAi) number 6 as described by Palmer and Strobeck[2] to assess intra-individual size variations between left and right-wing primordia or adult wings:

$$FAi = var\left[\frac{Ri - Li}{(Ri + Li)/2}\right], \quad (1)$$

where Ri and Li are the sizes of the right (R) and left (L) dissected discs or adult wings of the same individual. This FA index was chosen

because it normalizes left–right differences to average tissue size, and therefore prevents biases linked to experimental effects on average size (such as temperature changes or developmental time points). Figures represent the FAi ×10⁴. Only females (both for dissected discs and for adult wings) were analyzed.

## Statistics

For comparison of the means, two-tailed $t$-tests or ANOVA analysis (as indicated in the figure legends) were performed using GraphPad. No adjustments were done for multiple comparisons, given that only planned comparisons were performed and reported (we focused on few comparisons rather than every possible comparison, the choice of what to compare was part of the experimental design and we did not perform other comparisons than those planned). For FA data, we first assessed the normal distribution of asymmetry data $(2 \times (R - L)/(R + L))$ for each genotype using the Shapiro-Wilk test and inspecting the Normal Q-Q Plot, both provided by GraphPad. If all genotypes of an experiment showed normal distribution, a two-tailed F-test was used to compare variances (FAi values); otherwise Levene's test was used as indicated in the figure legends. In all cases, n values are indicated for each experiment in the corresponding figures or figure legends.

## Automated measurement of adult wings

In order to reduce the time and error in adult wing area quantification, we developed an automated deep learning-based segmentation technique. One hundred and fifty adult wings were acquired as indicated and images were stored as RGB.tif files. These pictures were used to manually generate binary masks of the wing blade, excluding the hinge region. Training data for the segmentation model was created using the RGB images as input and the binary mask as target. Images from different days of acquisition were used in the training dataset to create a high variation so that the trained model generalized well to images it had not seen before. Image augmentation techniques such as random rotation and flips were also used to make the model more robust. A UNET model was trained with a three-channel input image and a single channel output mask of the wing blade. The network depth was 5, with a training patch size of 1024 by 768 and a kernel size of 7. The model was trained for 150 epochs. Minor mistakes in the segmentation due to the presence of bubbles or debris near the wing surface were corrected with a Napari-based correction tool. The output of the program was set to be right-wing area (R), left-wing area (L), R − L, R + L, and asymmetry $(2 \times (R - L)/(R + L))$. From the asymmetry data, the FA index was calculated as indicated. Links to the codes, details, and instructions are provided hereafter in the "code availability" section and as Supplementary information.

## Immunostainings of larval tissues

Tissues dissected from both male and female larvae or pupae in 1× PBS at the indicated stages were fixed in 4% formaldehyde (ThermoScientific, #28908) in PBS for 30 min at room temperature, washed in PBS containing 0.3 % Triton-X-100 (PBT), blocked in PBT containing 2% BSA and incubated overnight with primary antibodies at 4 °C. The next day, tissues were washed, blocked again, and incubated with a 1/200 dilution of Alexa Fluor™ Plus 555 Phalloidin (ThermoFisher, Ref # A30106, Lot # 2420630) and/or secondary antibodies at 1/250 dilution for 2 h at room temperature. Secondary antibodies used were: Goat anti-Mouse IgG (H+L) Highly Cross-Adsorbed Secondary Antibody, Alexa Fluor™ 546 (Invitrogen, # A-11030, Lot # 2026145); Goat anti-Mouse IgG (H+L) Highly Cross-Adsorbed Secondary Antibody, Alexa Fluor™ Plus 555 (Invitrogen, # A32727, Lot # TE266003); Goat anti-Mouse IgG (H+L) Highly Cross-Adsorbed Secondary Antibody, Alexa Fluor™ Plus 647 (Invitrogen, # A32728, Lot # WE322197) and Goat anti-Chicken IgY (H+L) Secondary Antibody, Alexa Fluor™ 488 (Invitrogen, # A-11039, Lot # 2304258). Samples were mounted in Vectashield (Vector Labs, #H-1000-10) or SlowFade Diamond with DAPI (ThermoFisher, #S36964). Fluorescence images were acquired using a Zeiss LSM900 Inverted Laser Scanning Confocal Microscope, using a 40×/1,4 OIL DICII PL APO (UV) VIS-IR (420762-9800) or a 63×/ 1,4 OIL DICII PL APO (420782-9900) objective and a Zen microscopy software interface (Zeiss). Controls and experimental conditions images were equally processed using Fiji.

The following primary antibodies were used: chicken anti-GFP, 1/ 10,000 (Abcam, Ref # ab290, Lot # GR236651-12), mouse anti-FasIII, 1/50 (Developmental Studies Hybridoma Bank, Ref # 7G10), mouse anti-PH3, 1/200 (Cell Signaling, Ref # 9706) mouse anti-beta-galactosidase, 1/200 (Promega, Ref # Z3781, Lot #18637303). The epidermis of male and female WPP was dissected following fillet preparation protocols described in ref. 33: larvae and pupae of the required stages were rinsed with PBS to remove food debris and then placed with the dorsal side up in a dissection chamber, a 35-mm Petri dish with Sylgard 184 (SigmaAldrich, # 761036). One minutien pin (0.1 mm diameter stainless steel, Austerlitz Insect Pins) was pierced through the anterior of the larva (below the mouthparts) into the Sylgard, with the help of a pair of forceps to hold the animal and the pin. A second pin was pierced in the posterior of the animal, after extending its body. A drop of PBS was used to cover the animal. Using fine forceps, the posterior cuticle close to the pin was pinched to assist an initial cut with dissection scissors. This cut was pursued from posterior to anterior, to open the animal along the center. Two additional cuts were performed along the left side of the animal at the anterior and posterior ends, and then the left side was extended and secured with two extra pins. The same was done to extend and secure the right side. All tissues were carefully removed, except for the epidermis attached to the cuticle, and the preparation was fixed as described above by pouring fixative directly into the dissection chamber. For immunostaining and image acquisition, the fixed fillet was taken from the chamber and the same procedure described above for other tissues was applied.

## Clonal analysis

Crosses of the *hs-flp; act-FRT-STOP-FRT-GAL4,tub-Gal80ts; UAS-RFP* line with a *UAS-EcRDN; dilp8-GFP* line were performed in tubes with fly food and left at 25 °C. At L1 stage, a 30 min heat-shock was performed in a 42 °C water bath, using a Corio C model immersion circulator (Julabo, # T060100), to provoke a random flip-out activation of GAL4 expression. After the heat-shock, tubes were immediately transferred to 18 °C to repress GAL4 activity and avoid deleterious effects of EcRDN over-expression. At the L3w stage, larvae were shifted to 29 °C for one day to allow maximum activation of EcRDN expression and samples from females were dissected at the WPP stage.

## Quantitative RT-PCR

Male and female larvae or pupae were collected at the indicated stages. Whole animals or dissected tissues were frozen in liquid nitrogen. Total RNA was extracted using a RNeasy Lipid Tissue Mini Kit (Qiagen, # 74804) for whole larvae samples, or a QIAGEN RNeasy Micro Kit (for dissected wing discs) according to the manufacturer's protocol. Samples were handled by a QIAcube instrument (Qiagen, # 9002864) after homogenization with the Tissue Lyser II (Qiagen, #85300). RNA samples (2–3 μg per reaction) were treated with DNase when necessary and reverse-transcribed using SuperScript II reverse transcriptase (ThermoFisher, #18064022), and the generated cDNAs were used for real-time PCR (StepOne Plus, Applied Biosystems) using Power SYBR Green PCR mastermix (ThermoFisher, #A25741). Samples were normalized to *rp49* and fold changes were calculated using the ΔΔCt method; P values are the result of t-tests or ANOVA tests provided by Graphpad. At least three separate biological samples (5–10 animals each) were collected for each experiment and triplicate measurements were performed. The list of primers used is provided in Supplementary Table 1.

## Ecdysteroids extraction and quantification

For ecdysteroids extraction, 6–10 whole female and male animals at the indicated stages were collected for each biological replicate in 2 ml Eppendorf tubes, frozen in liquid nitrogen, and stored at −80 °C. Samples were homogenized using a metal bead and Tissue Lyser II (Qiagen, #85300) in 0.3 ml of methanol (SigmaAldrich, #322415), centrifuged at $18,000 \times g$ for 5 min at room temperature (RT) and the supernatant was transferred to a new tube. 0.3 ml of methanol was added, and samples were mixed using Vortex. This procedure was repeated using 0.3 ml of ethanol, so that the pooled samples contained a total volume of 0.9 ml, and were stored at −80 °C.

For quantification, the extracted samples were centrifuged at $18,000 \times g$ for 5 min (RT) to remove any remaining debris and divided into two tubes to generate technical replicates. The cleared samples were evaporated using a Speedvac centrifuge equipped with a cold trap. The following steps were performed using the 20-Hydroxyecdysone ELISA kit (Bertin Bioreagent, #A05120), with the following modifications: After evaporation, the precipitate was re-dissolved in 200 µl of ELISA buffer (EIA Buffer). It was critical to aid the re-dissolution of the precipitate by scraping it with a pestle and by vigorous vortexing, until no more was visible on the walls of the tubes. The ELISA plates were loaded with samples and a standard curve as indicated by the manufacturer, incubated overnight at 4 °C and read with a Microplate Reader (Tecan Sunrise) at 405 nm. Data analysis was performed as indicated in the Bertin 20-Hydroxyecdysone ELISA kit manual.

## Developmental timing measurement

Crosses were incubated at 25 °C and eggs were collected every 4 h on plates made of 2% agar and 2% sucrose in PBS. 22 h after the end of the 4h-oviposition, synchronized L1-stage animals were transferred from the agar plates to fly food and kept at 25 °C. Five groups of 30–50 larvae for each genotype were analyzed and the number of new pupae was scored every 2 h until all larvae pupariated.

## Reporting summary

Further information on research design is available in the Nature Research Reporting Summary linked to this article.

# Data availability

Source data are provided with this paper.

# Code availability

The codes for automated segmentation of whole wing area and quantification of fluctuating asymmetry are available in the Zenodo repository (https://doi.org/10.5281/zenodo.7026011), as well as tools for visualizing and correcting the segmentation results (https://doi.org/10.5281/zenodo.7025439). Detailed instructions and a demo folder are provided as Supplementary Data 1.

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

## Acknowledgements

We thank David Lubensky, Ojan Khatib-Damavandi, Lorette Noiret, and Marie-Anne Félix for insightful discussions; Paula Santa Bárbara-Ruiz for help with the experiments and members of the laboratory for discussions and comments on the manuscript; Cyril Kana Tepakbong for help with wing image segmentation codes; Julio Lopes Sampaio for help with ecdysteroids extractions; the Bloomington Stock Center and the Vienna *Drosophila* RNAi Center for fly stocks; the PICT-IBiSA@BDD light-microscopy facility of Institut Curie. This work was supported by Institut Curie, CNRS, INSERM, FRM, European Research Council (Advanced Grant no. 694677 to P.L.), HFSP grant no. RGP0031/2020 to D. Lubensky and P.L., Labex DEEP program (ANR-11-LABX-0044, ANR-10-IDEX-0001-02), PSL (Ph.D. fellowship to K.E.M.) and the Marie Sklodowska-Curie Actions (fellowship no. 897309 to D.B-O).

## Author contributions

Conceptualization and methodology: D.B-O., V.K., D.M., P.L., L.B. Investigations: D.B-O., K.E.M, F.B., L.V., D.S.A., J.C., S.N., L.B. Writing: D.B-O., D.M., P.L., L.B.

## Competing interests

The authors declare no competing interests.
