## [Peer Review File · Nature Communications]

A Dilp8-dependent time window ensures tissue size adjustment in *Drosophila*REVIEWER COMMENTS

Reviewer #1 (Remarks to the Author):

Comments

How organs achieve its final size and which are the mechanisms ensuring bilateral symmetry are important questions in developmental biology thought poorly understood. In *Drosophila*, null mutations for the relaxin-like hormone Dilp8 were previously found to decrease developmental stability, assessed by measuring fluctuating asymmetry index (FAi) between left and right wings. In this article, Boulan et al investigated the role of Dilp8 in tissue size adjustment during *Drosophila* wing development and found that ecdysone-dependent Dilp8 expression in the epidermis is crucial to reduce L-R wing asymmetry.

The article is very clear in showing that Dilp8, produced in the epidermis at larval-to-pupal (L-P) transition, is required for reducing FAi. Similarly, data is convincing in that Ecdysone signaling is required for both expressing Dilp8 (at this time point) and reducing wing asymmetries. Even though the findings are consistent with Ecdysone>Dilp8 defining a time window for wing size adjustment during development, the nature of the size adjustment mechanism and how Dilp8 controls this process remains unknown. Also, ecdysone's role acting downstream of Dilp8 and contributing to organ size adjustment seems preliminary.

Major points:

1- The authors showed that Dilp8 produced in epidermal cells is required for establishing a "time window" important for reducing organ asymmetries. However, it is not determine how this "time window/process" works.

a- Is there a time window at the end of wing imaginal disc growth period in which undergrowth disc catch up?

b- How does the wing disc grow during this stage (is it possible to show the grow trajectory for each individual disc from 96hAED to 7hAPF?)?

c- Is there additional rounds of cell division?

d- How are these or other cellular processes related to wing disc growth affected in dilp8 mutant animals?

e- Is wing size asymmetry the only parameter affected in dilp8^{-/-} or the absolute size is also affected (wings are smaller or bigger?)?

2- The authors showed that reducing Dilp8 expression has an impact on ecdysone levels at the L-P transition and on the expression of EcR-target genes in wing imaginal discs. Based on these findings the authors proposed a reciprocal feedback between the two hormones play a role in controlling organ size precision. I believe more experiments should be done to support this conclusion; i.e.: is the increased ecdysone signaling observed in *dilp8*^{-/-} wing disc cell-autonomously reducing growth?. Is this a direct consequence of the different ecdysone levels or of a premature peak of ecdysone? Is it possible to rescued or exacerbate FAi by playing with ecdysone levels and most important by modulating ecdysone signaling specifically in wing discs?

3- A recent paper identified a role of Dilp8-Lgr3 pathway in controlling pupariating behavior and puparium morphogenesis (Heredia et al 2020). They showed that ecdysone-dependent Dilp8 expression in epidermal cells signals to a cluster of 6 Lgr3-positive interneurons located in the VNC to promote pupariation. These results open up the possibility that Dilp8 produced at the epidermis controls wing disc growth acting via a different neuronal circuit than the one modulating ecdysone production. I believe that a further characterization of the subpopulation of Lgr3-positive neurons involved by using the different available cis-regulatory module (CRM)-Gal4 lines would be required to strengthen the conclusion of the paper.

4- It seems that there are others, *dilp8*-independent mechanisms, which contribute to reducing L-R wing asymmetry. Between 96-114 hs during larval development there is a clear reduction of FAi (Fig. 1) and also between pupae and adult wings (FAi in *dilp8* mutant pupal wings >20 vs FAi in *E22C>dilp8RNAi* adult wings = 2). If this is correct it would be important to more clearly discuss this point to have a better picture of the relative contribution of Dilp8-dependent mechanism in ensuring organ size symmetry.

Minor points,

- Line 260: misspelling "through".

Reviewer #2 (Remarks to the Author):

Boulan et al address the important question of how a bilaterally symmetrical organism achieves precision matching the size of corresponding appendages on the left right sides. They first provide quantitative data on the size of ing discs at different stages of larval development. They do this in both *dilp8* heterozygotes and homozygotes and make two conclusions: (1) The size matching occurs mainly during a specific interval around the time of pupation rather than throughout development and (2) That *dilp8* functions in this process during this time window. The authors then show that *dilp8* is expressed in

the cells of the epidermis, that expression peaks around that time, and that reducing epidermal dilp8 increases fluctuating asymmetry. The epidermal dilp8 expression requires ecdysone receptor function, but not the activity of factors that promote dilp8 expression in discs following damage. They reason that if Dilp8 acts via the same mechanisms that it does following disc damage, that it should reduce ecdysone levels. They do find that reducing dilp8 made by the epidermis increases ecdysone levels and increases expression of ecdysone target genes, They conclude that the correct level of ecdysone and ecdysone target gene expression is necessary for matching the size of the two imaginal discs.

Overall this study makes an important contribution to our understanding of organ size regulation by quantifying the extent of fluctuating asymmetry, by showing that dilp8 is made by the epidermis and that dilp8 regulates some ecdysone-dependent mechanism at the larval/pupal transition to allow the two wing discs to match each other's size.

I have two main issues with the conclusions reached by the authors and feel that these issues need to be addressed adequately before publication.

1. The authors conclude that there is a "major adjustment step" at the larval-pupal transition and contrast this with the development of the inner ear in zebrafish where the process is distributed through development. I am not convinced that the data support this conclusion. The authors never show any data that show when pupariation occurs in their system – they need to do this (I am guessing that it is at 120 hr based on figure 1C). If so the decline in fluctuating asymmetry in the wild type control (the dilp8 heterozygote) between 96 and 114 hr (18 hr) is comparable to that between 114 hr and 127 hr (7 hr APF) which is 13 hr duration. To me, this is as consistent with a steady decline in FA as it is with an "adjustment step". It is clear from the data that the later decline in FA is dilp8 dependent, but it could still be a gradual process regulated by two different mechanisms where some other mechanism is more important early on and dilp8 is more important later on.

2. If dilp 8 expression in the epidermis does not occur either because of RNAi or in the dilp8 mutant, ecdysone levels are elevated. I would expect that this might accelerate development and cause the larvae to pupate sooner. This might truncate the adjustment phase and stop the decrease in FA that occurs after this point. This might mean that dilp8 might not specifically regulate FA but might simply ensure that the larval phase is long enough for the adjustment to occur. The authors need to carefully compare developmental timing when dilp8 levels are normal or are reduced to distinguish between an indirect effect via developmental timing and some kind of direct effect of dilp8 or size matching. Do other genetic manipulations that accelerate pupariation or delay pupariation have effects on FA that might point to the simple conclusion that the longer the larval phase, the less the FA?

Reviewer #3 (Remarks to the Author):

Boulan et al. show that Dilp8 controls organ size adjustment in the *Drosophila* wing discs in a cross-organ manner. The authors demonstrate that deletion and depletion of *dilp8* compromises wing disc size adjustment at WPP and that *dilp8* is selectively expressed at WPP in the epidermis. Epidermal RNAi-driven depletion of *dilp8* increases organ asymmetry. Epidermal silencing of EcR inhibits *dilp8* gene expression and thus increases organ asymmetry, showing that ecdysone signaling is necessary to induce Dilp8 expression in the epidermis, which leads to organ size adjustment in the wing disc in a cross-organ manner. The work is convincing, data quality and presentation is very good. The manuscript is well-written and concise.

The finding that mutation of Dilp8 leads to increased organ asymmetry has already been shown. The authors have previously demonstrated that *dilp8* expression plays a critical role in limiting developmental variability (Boone 2016, NatComm). Now, the authors assessed wing disc pairs at larval and early pupal development to show that wing disc asymmetry is corrected at WPP in a *dilp8*-dependent manner. Thereby, ecdysone-signaling induces Dilp8 expression in the epidermis within a sharp time-window, which controls size adjustment of the wing disc. These findings are in principle very interesting, but do not extend present knowledge enough to be suitable for publication in Nature Communications. Mechanistic insights into how *dilp8* expression in the epidermis does control wing disc size would be essential, in particular as the principle finding that *dilp8* ensures organ size adjustment has already been shown.

Major:

1. The authors need to provide some insights into how *dilp8* expression in the epidermis controls size adjustment in the wing disc. So far, it remains elusive how *dilp8* expression in the epidermis translates into alterations of gene expression in the wing disc to control organ size.

4. Is Dilp8 expression sufficient to induce organ size adjustment also at other developmental time points? What about too early expression of *dilp8* during development? Would for instance Dilp8 overexpression in the epidermis at early larval development already adjust variability in the wing discs as seen for the *dilp8* peak at WPP?

2. Does overexpression of *dilp8* in the wing disc instead of the epidermis also adjust organ size? Or is overexpression in the epidermis necessary to control wing disc size?

3. The authors suggest a feedback loop between ecdysone and dilp8 in which EcR signaling in the epidermis drives dilp8 expression, and dilp8 affects ecdysone levels (per animal) and expression of EcR target genes in the wing disc. Can dilp8 expression also induce organ size adjustment when ecdysone signaling in the receiving organ/wing disc is compromised, e.g. EcR-DN or EcR-RNAi in wing discs instead of epidermis to leave expression of Dilp8 in the epidermis intact?

5. Inactivation of EcR-signaling in the epidermis inhibits dilp8 expression at WPP. Is this a direct effect of EcR or are additional EcR-responsive genes involved? How is dilp8-expression achieved? The authors could for instance test their hypothesis that the selective expression of dilp8 in the epidermis involves the EcR-B2 isoform.

Minor:

- Fig. S2C is this a correct overlay? Cell borders seem off in respect to the signal of the Dilp8-GFP reporter.

- A model would be helpful.

RESPONSE TO REVIEWERS' COMMENTS

Reviewer #1:

How organs achieve its final size and which are the mechanisms ensuring bilateral symmetry are important questions in developmental biology thought poorly understood. In *Drosophila*, null mutations for the relaxin-like hormone Dilp8 were previously found to decrease developmental stability, assessed by measuring fluctuating asymmetry index (FAi) between left and right wings. In this article, Boulan et al investigated the role of Dilp8 in tissue size adjustment during *Drosophila* wing development and found that ecdysone-dependent Dilp8 expression in the epidermis is crucial to reduce L-R wing asymmetry.

The article is very clear in showing that Dilp8, produced in the epidermis at larval-to-pupal (L-P) transition, is required for reducing FAi. Similarly, data is convincing in that Ecdysone signaling is required for both expressing Dilp8 (at this time point) and reducing wing asymmetries.

We thank the Reviewer for the positive comments.

Even though the findings are consistent with Ecdysone>Dilp8 defining a time window for wing size adjustment during development, the nature of the size adjustment mechanism and how Dilp8 controls this process remains unknown. Also, ecdysone's role acting downstream of Dilp8 and contributing to organ size adjustment seems preliminary.

Major points:

1- The authors showed that Dilp8 produced in epidermal cells is required for establishing a "time window" important for reducing organ asymmetries. However, it is not determine how this "time window/process" works.

a- Is there a time window at the end of wing imaginal disc growth period in which undergrowth disc catch up?

We show in Fig. 1b,c that the volume difference between L and R wing discs drastically reduces between 114h AED and 7hr APF. Whether this is due to a catch up growth of the small disc, or a growth deceleration of the large disc is a puzzling question. We tried to address this experimentally using live microscopy. Unfortunately, following disc growth on both sides in living (and moving) larvae turned out to be technically challenging. Additionally, it is difficult to define a standard growth trajectory for a given tissue in the larva. Indeed, individual variation in body size would necessitate using allometric comparisons with other body parts, which is not precise enough given the small L/R variation observed. It is therefore not possible for us to determine whether a disc is overgrown or undergrown compared to its expected size at a given larval or prepupal stage.

b- How does the wing disc grow during this stage (is it possible to show the grow trajectory for each individual disc from 96hAED to 7hAPF?)?

Unfortunately, we lack precision in our analysis of the growth trajectories on fixed tissues during the transition between larva and prepupa. Again, live -microscopy might help to follow individual growth trajectories, but is out of reach for the moment. Our general estimate is that discs double their volume between the end of larval development and the first hours of pre-pupa, indicative of a very active growth during the adjustment window. Interestingly this period is marked by an intense expansion of the tissue (see our Fig. 1a). A time window has also been described in zebrafish development during which bilateral inner ears and somites adjust their size (Green et al. Dev Dyn. 2017, Naganathan et al. Nature 2022). This suggests that tissue expansion during specific periods of development could participate in the process of size adjustment.

c- Is there additional rounds of cell division? How are these or other cellular processes related to wing disc growth affected in dilp8 mutant animals?

We now provide additional data clarifying this point in our new **Fig. 5d**. We measured the number of cell divisions (PH3-positive cells) in developing discs at 3 time points during the adjustment window (0hr APF, 2hr APF, 4hr APF). As previously described (Guo et al. Biol. Open 2016), we observe a gradual proliferation arrest early after WPP stage in control discs. Interestingly, this proliferation arrest is accelerated in *dilp8* mutants, therefore shortening the time during which disc cells continue dividing. This is in line with the increase in ecdysone accumulation at 0hr APF observed in *dilp8* mutants (see **Fig. 5a**), since high ecdysone titers induce a transcriptional inhibition of the mitotic inducer String/Cdc25 during this period (Guo et al. 2016).

e- Is wing size asymmetry the only parameter affected in *dilp8*^{-/-} or the absolute size is also affected (wings are smaller or bigger)?

Indeed, the absolute size of *dilp8* mutant wings is slightly reduced compared to control (see new **Fig S1b**, and refs Garelli et al. 2012, Boone et al. 2015).

2- The authors showed that reducing Dilp8 expression has an impact on ecdysone levels at the L-P transition and on the expression of EcR-target genes in wing imaginal discs. Based on these findings the authors proposed a reciprocal feedback between the two hormones play a role in controlling organ size precision. I believe more experiments should be done to support this conclusion; i.e.: is the increased ecdysone signaling observed in *dilp8*^{-/-} wing disc cell-autonomously reducing growth? We now provide a rescue experiment in which the increased FA observed in a *dilp8*^{KO/ag54} null mutant is rescued by the local inhibition of Ecdysone import in disc cells (*nub>Ecl-RNAi*) (new **Fig. 5c**). Therefore, the increased ecdysone signaling observed in *dilp8* mutant wing discs at the L/P transition is responsible for the FA phenotype. As described in our response to point 1c, we show that in a *dilp8* mutant background, increased levels of Ecdysone at the L/P transition are associated with a faster decrease in cell proliferation during early prepupa (new **Fig. 5d**). We propose that this reduced proliferation could participate in the increased FA observed in these conditions.

Is this a direct consequence of the different ecdysone levels or of a premature peak of ecdysone?

Three lines of evidence indicate that the Dilp8 phenotype is not a consequence of an early ecdysone peak. (i) Dilp8 is expressed very sharply at the white prepupal stage, after the raise of ecdysone is established. (ii) There are no significant differences in Ecdysteroids titers between controls and *dilp8*^{KO/KO} null mutants at wandering L3 stage (**Fig. 5a**). (iii) We now provide the data showing that there is no difference in pupariation time in three Dilp8 loss-of-function situations that induce FA (*dilp8*^{KO/KO}, *E22C>dilp8-RNAi* or *Eip71CD>dilp8-RNAi*) compared to controls (new **Fig. S4a-c** and ref. Garelli et al. 2012). These data are consistent with the notion that Dilp8 acts right after pupariation and therefore that the ecdysone peak is not premature.

Is it possible to rescued or exacerbate FAi by playing with ecdysone levels and most important by modulating ecdysone signaling specifically in wing discs?

We now provide new experimental evidence that confirm this point.

- the increased FA observed in a *dilp8*^{KO/ag54} mutant is rescued by the local inhibition of Ecdysone import in disc cells (*nub>Ecl-RNAi*) (new **Fig. 5c**).

- the increased FA observed in an *act>dilp8-RNAi* context is rescued by reducing PTTH expression (*PTTH-RNAi*), which limits Ecdysone production (new **Fig. S4f**).

3- A recent paper identified a role of Dilp8-Lgr3 pathway in controlling pupariating behavior and puparium morphogenesis (Heredia et al 2020). They showed that ecdysone-dependent Dilp8 expression in epidermal cells signals to a cluster of 6 Lgr3-positive interneurons located in the VNC to promote pupariation. These results open up the possibility that Dilp8 produced at the epidermis controls wing disc growth acting via a different neuronal circuit that the one modulating ecdysone

production. I believe that a further characterization of the subpopulation of Lgr3-positive neurons involved by using the different available cis-regulatory module (CRM)-Gal4 lines would be required to strengthen the conclusion of the paper.

We indeed tested the knockdown of Lgr3 in these 6 interneurons (*18A01>lgr3-RNAi*) and did not observe FA. We also confirmed in the same experiment that Lgr3 silencing in the 19B09 neurons, which mediate the Dilp8-induced developmental delay, increases FA (new. Fig. S4e).

4- It seems that there are others, dilp8-independent mechanisms, which contribute to reducing L-R wing asymmetry. Between 96-114 hs during larval development there is a clear reduction of FAi (Fig. 1) and also between pupae and adult wings (FAi in dilp8 mutant pupal wings > 20 vs FAi in E22C>dilp8RNAi adult wings = 2). If this is correct it would be important to more clearly discuss this point to have a better picture of the relative contribution of Dilp8-dependent mechanism in ensuring organ size symmetry.

We thank the Reviewer for this remark, which is shared by Reviewer 2. We have re-analyzed our data concerning developmental FA. Although there is a trend for a reduction of FA between 96hr and 114hr AEL, this is not significant (see new Fig. 1 b,c). We apologize to the Reviewer for not having provided this statistical analysis earlier.

Comparing FA in pupal and adult wings turned out to be difficult. Indeed, while a volume is measured in pupal tissues, a surface is quantified in adult wings. Whether other size adjustments take place later during pupal development is a very interesting topic, which is the subject of our current research. We corrected the text in the discussion to mention these remarks.

Minor points,

- Line 260: misspelling "through".

We thank the Reviewer and have modified the text accordingly.

Reviewer #2:

Boulan et al address the important question of how a bilaterally symmetrical organism achieves precision matching the size of corresponding appendages on the left right sides. They first provide quantitative data on the size of ing discs at different stages of larval development. They do this in both dilp8 heterozygotes and homozygotes and make two conclusions: (1) The size matching occurs mainly during a specific interval around the time of pupation rather than throughout development and (2) That dilp8 functions in this process during this time window. The authors then show that dilp8 is expressed in the cells of the epidermis, that expression peaks around that time, and that reducing epidermal dilp8 increases fluctuating asymmetry. The epidermal dilp8 expression requires ecdysone receptor function, but not the activity of factors that promote dilp8 expression in discs following damage. They reason that if Dilp8 acts via the same mechanisms that it does following disc damage, that it should reduce ecdysone levels. They do find that reducing dilp8 made by the epidermis increases ecdysone levels and increases expression of ecdysone target genes, They conclude that the correct level of ecdysone and ecdysone target gene expression is necessary for matching the size of the two imaginal discs.

Overall this study makes an important contribution to our understanding of organ size regulation by quantifying the extent of fluctuating asymmetry, by showing that dilp8 is made by the epidermis and that dilp8 regulates some ecdysone-dependent mechanism at the larval/pupal transition to allow the two wing discs to match each other's size.

I have two main issues with the conclusions reached by the authors and feel that these issues need to be addressed adequately before publication.

1. The authors conclude that there is a “major adjustment step” at the larval-pupal transition and contrast this with the development of the inner ear in zebrafish where the process is distributed through development. I am not convinced that the data support this conclusion. The authors never show any data that show when pupariation occurs in their system – they need to do this (I am guessing that it is at 120 hr based on figure 1C).

We have measured pupariation times and added this piece of data to the new version of our manuscript (new Fig. S4a-c). We find that control animals pupariate at 138h AED, slightly later than what is described in the literature, which is due to a rather low concentration of yeast in the fly food. We observe no difference in pupariation time in *dilp8^{KO/KO}*, *E22C>dilp8-RNAi* or *Eip71CD>dilp8-RNAi* compared to controls (new Fig. S4a-c and ref. Garelli et al. 2012).

If so the decline in fluctuating asymmetry in the wild type control (the *dilp8* heterozygote) between 96 and 114 hr (18 hr) is comparable to that between 114 hr and 127 hr (7 hr APF) which is 13 hr duration. To me, this is as consistent with a steady decline in FA as it is with an “adjustment step”. It is clear from the data that the later decline in FA is *dilp8* dependent, but it could still be a gradual process regulated by two different mechanisms where some other mechanism is more important early on and *dilp8* is more important later on.

We have re-analyzed our data concerning developmental FA. Although there is a trend for a reduction of FA between 96hr and 114hr AEL, this is not significant. We apologize to the Reviewer for not having provided this statistical analysis earlier. The fact that the size adjustment around L/P transition is linked to a sharp expression of *dilp8* at WPP suggests that it has a defined start during early prepupa. We now provide new data showing that the disc proliferation arrest that gradually takes place during early prepupa (0-4hr APF), is accelerated in *dilp8* mutants (new Fig. 5d), concomitant with an increase in ecdysone accumulation. Therefore, several hormonal and cellular events participate in the adjustment posterior to the *Dilp8* expression peak, which could make this phase of adjustment spanning over a yet undefined period of prepupal development. We corrected the text accordingly.

2. If *dilp8* expression in the epidermis does not occur either because of RNAi or in the *dilp8* mutant, ecdysone levels are elevated. I would expect that this might accelerate development and cause the larvae to pupate sooner. This might truncate the adjustment phase and stop the decrease in FA that occurs after this point. This might mean that *dilp8* might not specifically regulate FA but might simply ensure that the larval phase is long enough for the adjustment to occur. The authors need to carefully compare developmental timing when *dilp8* levels are normal or are reduced to distinguish between an indirect effect via developmental timing and some kind of direct effect of *dilp8* or size matching.

We thank the Reviewer for this important remark. We bring new data showing that *dilp8* mutant larvae, as well as larvae with *dilp8* silencing in the epidermis, show unperturbed pupariation timing (new Fig. S4a-c) (see also Garelli et al. 2012 with a different *dilp8* mutant allele). In addition, *dilp8* mutant larvae show no detectable difference in ecdysone levels before pupariation (Fig. 5a). According to these observations, the high FA observed in *dilp8* mutants is not due to a shortening of the larval period.

Do other genetic manipulations that accelerate pupariation or delay pupariation have effects on FA that might point to the simple conclusion that the longer the larval phase, the less the FA?

As stated above, we clearly show that there is no pupariation time difference between controls and the various *dilp8* loss-of-function conditions. However, these animals present high FA, indicating that there is not link between high FA and developmental time.

As asked by the Reviewer, we performed manipulations to directly assess the effect of the duration of the larval period on FA. Over-expression of *PTTH* (*tub > PTTH*) accelerates the L/P transition by 12hr and increases FA, as shown below:

However, this genetic background also leads to a general increase in ecdysone levels, which could explain the high FA phenotype, as demonstrated in the case of the *dilp8* mutant. Given this ambiguity in its interpretation, we chose not to include this piece of data in our manuscript.

Conversely, we present the analysis of a *ptth* null mutant with delayed pupariation (Shimell et al, 2018), but normal FA (new Fig. S4d). A similar result is observed upon ubiquitous downregulation of *ptth* (*act > ptth^{RNAi}*, new Fig. S4f). Therefore, genetic manipulations that increase the duration of the larval phase do not reduce L-R variations.

Altogether, this indicates that developmental events downstream of *dilp8* expression at WPP are required for proper L-R size adjustment, rather than the duration of the larval period.

Reviewer #3:

Boulan et al. show that Dilp8 controls organ size adjustment in the Drosophila wing discs in a cross-organ manner. The authors demonstrate that deletion and depletion of *dilp8* compromises wing disc size adjustment at WPP and that *dilp8* is selectively expressed at WPP in the epidermis. Epidermal RNAi-driven depletion of *dilp8* increases organ asymmetry. Epidermal silencing of EcR inhibits *dilp8* gene expression and thus increases organ asymmetry, showing that ecdysone signaling is necessary to induce Dilp8 expression in the epidermis, which leads to organ size adjustment in the wing disc in a cross-organ manner. The work is convincing, data quality and presentation is very good. The manuscript is well-written and concise.

The finding that mutation of Dilp8 leads to increased organ asymmetry has already been shown. The authors have previously demonstrated that *dilp8* expression plays a critical role in limiting developmental variation (Boone 2016, NatComm). Now, the authors assessed wing disc pairs at larval and early pupal development to show that wing disc asymmetry is corrected at WPP in a *dilp8*-dependent manner. Thereby, ecdysone-signaling induces Dilp8 expression in the epidermis within a sharp time-window, which controls size adjustment of the wing disc. These findings are in principle very interesting, but do not extend present knowledge enough to be suitable for publication in Nature Communications. Mechanistic insights into how *dilp8* expression in the epidermis does control wing disc size would be essential, in particular as the principle finding that *dilp8* ensures organ size adjustment has already been shown.

We believe that this work constitutes an important step with the finding that size adjustment is controlled at a specific timepoint towards the end of the growth period through a time window mechanism depending on Dilp8. This contrasts with the alternative notion that precision is achieved through continuous and progressive adjustment during the growth phase. We are also very much

interested in exploring the mechanisms of tissue size adjustment and this is actually the topic of our current research. Thanks to the conclusions of the present study we can now focus in a rather narrow developmental window (0-7h APF) to closely evaluate the growth trajectories and morphogenetic events taking place, and study which one is crucial for tissue size adjustment. We provide here the insight that cell proliferation at this moment might be required, but an important number of experiments will be required to confirm this and/or alternative hypothesis and will be the focus of a future publication.

Major:

1. The authors need to provide some insights into how *dilp8* expression in the epidermis controls size adjustment in the wing disc. So far, it remains elusive how *dilp8* expression in the epidermis translates into alterations of gene expression in the wing disc to control organ size.

We have shown that in *dilp8* mutants there is indeed an alteration of the ecdysone-dependent gene expression autonomously in the wing discs (**Fig 5b**). Additionally, we now show through a rescue experiment that the ecdysone-dependent response is indeed required in the wing discs for the increased FA phenotype of the *dilp8* mutant animals (new **Fig 5c**). Also, we now provide the result that proliferation dynamics is perturbed in *dilp8* mutant discs (see PH3 staining at 0, 2, 4hr APF in *wt* and *dilp8* mutants, new **Fig. 5d**). Our data indicate that the progressive proliferation arrest observed in prepupal discs is accelerated in *dilp8* mutants, potentially reducing the total number of cell divisions. This could affect the ability of discs to adjust their size. Consistent with this, *dilp8* mutant adult wings are slightly reduced in size (new **Fig S1b**). Altogether, these results indicate that *dilp8* expression in the epidermis is required to execute the proper ecdysone-dependent gene expression program in the wing discs and suggest that the amplitude of this program is crucial for adequate wing size adjustment.

2. Does overexpression of *dilp8* in the wing disc instead of the epidermis also adjust organ size? Or is overexpression in the epidermis necessary to control wing disc size?

We did not succeed in doing this ectopic rescue experiment for technical reasons (strong lethality observed after combining the *dilp8* mutation with other genetic modifications).

In another paradigm (disc growth impairment, see Boulan et al. 2019), Dilp8 is produced by the disc and acts remotely on the central system to control ecdysone production. This suggests that Dilp8 can be produced by another tissue than the epidermis to modulate ecdysone production. However, its role in organ size precision was not evaluated in this context. We show here that silencing *dilp8* in the disc during normal development does not induce FA, indicating that Dilp8 function is not needed in the disc for size adjustment (**Fig. S2a**).

We speculate on the reasons for Dilp8 being produced in the epidermis in our discussion.

3. The authors suggest a feedback loop between ecdysone and *dilp8* in which EcR signaling in the epidermis drives *dilp8* expression, and *dilp8* affects ecdysone levels (per animal) and expression of EcR target genes in the wing disc. Can *dilp8* expression also induce organ size adjustment when ecdysone signaling in the receiving organ/wing disc is compromised, e.g. EcR-DN or EcR-RNAi in wing discs instead of epidermis to leave expression of Dilp8 in the epidermis intact?

Indeed, we have performed the rescue experiment of a Dilp8 mutant using disc-specific downregulation of the Ecdysone Importer (*nub>Ecl-RNAi*, see new **Fig. 5c**).

4. Is Dilp8 expression sufficient to induce organ size adjustment also at other developmental time points? What about too early expression of *dilp8* during development? Would for instance Dilp8 overexpression in the epidermis at early larval development already adjust variation in the wing discs as seen for the *dilp8* peak at WPP?

We have tested these experiments, but they turned out to be uninformative. We observed that when Dilp8 is induced before pupariation it delays pupariation, therefore introducing a strong perturbation

during larval development. In addition, we repeatedly observed that overexpression of Dilp8 is sufficient to induce FA. Therefore, proper levels of Dilp8 are required for tissue size adjustment. We observed that Dilp8 is also expressed at a later time point during pupal development. Whether this is linked to organ size adjustment is currently under investigation.

5. Inactivation of EcR-signaling in the epidermis inhibits dilp8 expression at WPP. Is this a direct effect of EcR or are additional EcR-responsive genes involved? How is dilp8-expression achieved? The authors could for instance test their hypothesis that the selective expression of dilp8 in the epidermis involves the EcR-B2 isoform.

As asked by the Reviewer, we performed experiments over-expressing different dominant-negative EcR isoforms in the epidermis (*Eip71CD-Gal4*) to test their effect on Dilp8 expression at the WPP stage. Unfortunately, all these genetic backgrounds present high lethality at the L/P transition, and did not allow to measure *dilp8* expression levels. To circumvent this, we used an *Eip71CD-Gal4, tub-Gal80^{ts}* background to shorten over-expression times. While we succeeded at getting live WPP after a short 6hr over-expression, we couldn't observe an effect on *dilp8* expression, which could be explained by residual EcR function. Due to these technical difficulties, we were therefore unable to conclude for a specific requirement of EcR-B2 for *dilp8* expression.

Minor:

- Fig. S2C is this a correct overlay? Cell borders seem off in respect to the signal of the Dilp8-GFP reporter.

Indeed, the Reviewer is right. We have gone through many of our images and a similar effect is observed in some cells. As shown in the lateral view in **Fig 3b**, the plane of the nucleus does not necessarily coincide with the plane of Fas III staining. Since we are using maximum projections of a Z-stack to avoid the many out-of-focus wrinkles of the epidermis, it is expected that in the final projection some cell bodies can display a slight displacement compared to the cell borders marked by Fas III. However, it should be noted that the localization of cell borders is aligned with the localization of nuclei.

- A model would be helpful.

We now provide a model in **Fig. 5e**.

REVIEWERS' COMMENTS

Reviewer #1 (Remarks to the Author):

In this revised version of the article, Boulan et al have provided additional data that clarify some of my previous concerns: (i) they have included experiments showing that Dilp8 controls developmental precision by modulating ecdysone signalling in target tissues; (ii) they showed that Lgr3 silencing in the 19B09 neurons increases FA; they have provided evidences that the proliferation arrest that takes place early after WPP is accelerated in dilp8 mutants. Even though the authors were not able to provide more details concerning how the size adjustment mechanism actually works, I believe the article has been very much improved.

Reviewer #2 (Remarks to the Author):

The authors have adequately addressed all of my concerns.

This is now a very nice and clean story that presents the very novel finding that the epidermis functions as the source of a hormone that regulates the level of precision that can be achieved in determining the final size of an organ.

I have one very minor suggestion:

In line 239, the authors write: Noticeably, removing a Yki/Sd response element in the dilp8 promoter leads to a rather mild increase in developmental stability.

Please explain what you mean by "developmental stability".

Reviewer #3 (Remarks to the Author):

The additional data sets included by the authors in the course of the revision substantially strengthen and extend the findings and conclusions presented in the initial manuscript. Overall, the manuscript has been clearly improved. I do not have any remaining concerns and suggest publication of this study.

Response to Reviewers :

Reviewer #1 (Remarks to the Author):

In this revised version of the article, Boulan et al have provided additional data that clarify some of my previous concerns: (i) they have included experiments showing that Dilp8 controls developmental precision by modulating ecdysone signalling in target tissues; (ii) they showed that Lgr3 silencing in the 19B09 neurons increases FA; they have provided evidences that the proliferation arrest that takes place early after WPP is accelerated in dilp8 mutants. Even though the authors were not able to provide more details concerning how the size adjustment mechanism actually works, I believe the article has been very much improved.

We thank the Reviewer for the positive assessment.

Reviewer #2 (Remarks to the Author):

The authors have adequately addressed all of my concerns.

This is now a very nice and clean story that presents the very novel finding that the epidermis functions as the source of a hormone that regulates the level of precision that can be achieved in determining the final size of an organ.

I have one very minor suggestion:

In line 239, the authors write: Noticeably, removing a Yki/Sd response element in the dilp8 promoter leads to a rather mild increase in developmental stability.

Please explain what you mean by "developmental stability".

We thank the Reviewer for the positive assessment.

We apologize for the mistake. This sentence has now been corrected for "Noticeably, removing a Yki/Sd response element in the dilp8 promoter leads to a rather mild decrease in developmental stability".

Reviewer #3 (Remarks to the Author):

The additional data sets included by the authors in the course of the revision substantially strengthen and extend the findings and conclusions presented in the initial manuscript. Overall, the manuscript has been clearly improved. I do not have any remaining concerns and suggest publication of this study.

We thank the Reviewer for the positive assessment.